# Olive fruit fly and its obligate symbiont *Candidatus* Erwinia dacicola: Two new symbiont haplotypes in the Mediterranean basin

**Tânia Nobre** *

MED - Mediterranean Institute for Agriculture, Environment and Development, Instituto de Investigação e Formação Avançada, Universidade de Évora, Pólo da Mitra, Évora, Portugal

* tnobre@uevora.pt

## Abstract

The olive fruit fly, specialized to become monophagous during several life stages, remains the most important olive tree pest with high direct production losses, but also affecting the quality, composition, and inherent properties of the olives. Thought to have originated in Africa is nowadays present wherever olive groves are grown. The olive fruit fly evolved to harbor a vertically transmitted and obligate bacterial symbiont -*Candidatus* Erwinia dacicola-leading thus to a tight evolutionary history between olive tree, fruit fly and obligate, vertical transmitted symbiotic bacterium. Considering this linkage, the genetic diversity (at a 16S fragment) of this obligate symbiont was added in the understanding of the distribution pattern of the holobiont at nine locations throughout four countries in the Mediterranean Basin. This was complemented with mitochondrial (four mtDNA fragments) and nuclear (ten microsatellites) data of the host. We focused on the previously established Iberian cluster for the *B. oleae* structure and hypothesised that the Tunisian samples would fall into a differentiated cluster. From the host point of view, we were unable to confirm this hypothesis. Looking at the symbiont, however, two new 16S haplotypes were found exclusively in the populations from Tunisia. This finding is discussed in the frame of host-symbiont specificity and transmission mode. To understand olive fruit fly population diversity and dispersion, the dynamics of the symbiont also needs to be taken into consideration, as it enables the fly to, so efficiently and uniquely, exploit the olive fruit resource.

## Introduction

The olive tree was likely the first domesticated fruit tree, and although domestication probably began in the Eastern Mediterranean, selection for cultivars took place at several different independent locations [1–3]. Given time and appropriate conditions, the olive fruit fly evolved to explore this resource. This fruit fly, *Bactrocera oleae* (Rossi, 1790), has specialized to become monophagous and it remains the most important olive tree pest. Production losses are

**Data Availability Statement:** All relevant data are within the manuscript and Supporting information files.

**Funding:** This work is supported by National Funds through FCT - Foundation for Science and

Technology under the research project PTDC/ASP-PLA/30650/2017. The funders had no role in study design, data collection and analysis, decision to publish, or preparation of the manuscript.

**Competing interests:** The authors have declared that no competing interests exist.

estimated on an average of more than 15% yearly [4], and this fly has been responsible for losses of up to 80% of oil value and 100% of some table cultivars [5]. To cope with the olive-plant abundant secondary metabolites, particularly the defensive compound oleuropein (a bitter and otherwise toxic phenolic glycoside) [6], the olive fruit fly evolved to harbor a vertically transmitted and obligate bacterial symbiont—*Candidatus* Erwinia dacicola [7] leading thus to a tight evolutionary history between olive tree, fruit fly and obligate, vertical transmitted symbiotic bacterium.

The olive fruit fly is thought to have originated in Africa and then spread to the Mediterranean basin and South Central Asia [8], and it is nowadays present wherever olive groves are grown. Nardi and co-workers [9] presented the first comprehensive study on olive fly populations and found evidence of a population subdivision into three regional groups, Pakistan, Africa and Mediterranean plus America. Further, there is evidence of a separation between Iberian (Western Mediterranean) and Italic (Central Mediterranean) olive fly populations [10], with no clearly defined boundary [8]. Van Asch and co-workers [8] observed intermixing to extend at variable levels throughout their whole studied area, from Northwestern Italy to Portugal.

As an obligate symbiont, mechanisms that ensure vertical transmission have evolved and *Ca*. Erwinia dacicola is transmitted to the following generation at the oviposition [7,11,12] guaranteeing symbiont acquisition by the offspring but also host-symbiont specificity. When transmission is strictly vertical, host switching would not be expected, and co-evolution of host and symbiont lineages should be observed. For both symbiont and host, failure in vertical transmission has high costs, and is usually equivalent to death of the symbiont and significant reduction in host fitness or high mortality. Selection at these endosymbiotic bacteria likely results from selection imposed by the host and that from selection emerging in a symbiotic context but independent from the host [13]. Furthermore, the cyclic events of transmission -where a given number of symbionts are passed to the next generation- are likely to create strong bottlenecks affecting symbiont population dynamics. Not much is known on the diversity and structure of *ca*. Erwinia dacicola except that there is evidence of two lineages of this endosymbiont in Italian populations, based on 16S rRNA gene sequences [14].

Understanding how the pest population spreads is essential for its management. The close association of this monophagous fly species with its host, and its possibility of laying eggs throughout most of the year, allows for a fairly stable population and the rate of population change is mainly shaped by density-dependent population feedback [15]. Its spreading is intricately connected to the spreading of the plant host, as this relation seems the result of long-term co-evolution. In most areas where olives are grown commercially, the olive fruit fly can be found and have become a major pest. In North America was first detected in 1998 in California [16], and it has spread throughout the region, including in Central and South America. Nowadays it seems that the olive fruit fly can also be found well outside the Mediterranean basin in Central and South Africa, and not only in the Middle East but also in China and India; Australia seems to be the only area in the world, where olives are cultivated and well established for commercial purposes, which is still free of olive fruit fly [17].

Having as rationality this intricate tripartite relation between olive tree, olive fruit fly and *Ca*. Erwinia dacicola symbiont, this work aimed to add this obligate symbiont genetic diversity in the understanding of the distribution pattern of the holobiont at nine locations throughout four countries in the Mediterranean Basin. We focused on the Iberian cluster and on the hypothesis that the Tunisian samples would fall into a differentiated cluster due to the proximity to the Italic peninsula cluster. To date, and to the best of our knowledge, the genetic diversity present within the olive fly symbiont has only investigated for the Italian populations [14]. Therefore, the present study looked at a) haplotypes from four mitochondrial DNA fragments

**Table 1. Sampling locations and collection year (see acknowledgements for collectors' information).**

| Country | Location | Code | Coordinates | | Year |
|---|---|---|---|---|---|
| Portugal | Guimarães | GUI | 41.46N | 8.31W | 2016 |
| Portugal | Valverde | VAL | 38.86N | 7.27W | 2019 |
| Portugal | Lagos | LAG | 37.13N | 8.68W | 2016 |
| Spain | LaRoda | LRD | 39.09N | 2.19W | 2019 |
| Spain | Seville | SEV | 37.47N | 5.99W | 2019 |
| Spain | Almeria | ALM | 36.89N | 2.44W | 2016 |
| France | Montpellier | MON | 43.61N | 3.87E | 2017 |
| Tunisia | Bouficha | BOU | 36.30N | 10.45E | 2017 |
| Tunisia | Zarzis | ZAR | 33.50N | 11.12E | 2017 |

of the host fly, b) patterns of ten microsatellites of the host fly and c) 16S rRNA gene sequence diversity of *Ca.* E. dacicola. Albeit with a limited geographical and populational coverage, it was intended to look for population differentiation, gaining insights into gene flow and spreading. Overall rising temperatures, warmer winter minimum temperatures and changes in precipitation patterns that are likely linked to water shortages [18] are affecting the spreading of insects and their aggressiveness as pests in largely unknown ways [19]. These changes will lead to species specific dynamics [20–22], which likely will impact on the interaction of the olive and its obligate olive fruit fly.

## Methodology

### Data collection

Olive fruit flies were sampled at 9 locations (3 in Portugal, 3 in Spain, 1 in France and 2 in Tunisia; Table 1), either directly, with the permission of the owners, or made available by colleagues and/or landowners (see acknowledgements). Sampling did not involve endangered or protected species. Olives were collected and stored in plastic boxes, with emerging pupae and adults being gathered up to 12 specimens per location. Individuals were stored at -20 ˚C in 70% ethanol until DNA extraction. Individuals were allowed to dry on filter paper prior to DNA extraction. DNA from whole body tissue was extracted following extraction protocols [23] using CTAB extraction buffer after being ground up with a plastic pestle. Proteins were removed with 24: 1 isoamylalcohol: chloroform, and DNA precipitated with isopropanol. DNA extracts were eluted in 50 μl of sterile water. All extraction products were stored at -20 ˚C and later used directly in the PCRs.

### Olive fruit fly mtDNA sequences

Four highly variable sections of mtDNA were amplified and sequenced. These were amongst the five sections selected by van Asch and co-workers [10], based on the number of polymorphisms previously described in a Mediterranean region dataset [24]. These include both tRNA-Leu genes and segments of the ND2, ND4, COX1 and COX2 and genes, ca. 24% of the complete mitochondrial genome of *B. oleae* [10] (see S1 Table for primer information). PCR reactions were conducted using 1 μl of the extracted DNA in a standard 25 μl reaction, with 0.5 pmol/μl of each primer, 1.5 mM $MgCl^2$, 0.5 mM dNTPs and 0.04 U/ml Taq DNA polymerase. The touch-down cycle protocol involved two-phases; 1) initial denaturation at 94 ˚C for 5 min, followed by 10 cycles of denaturation at 94 ˚C for 30 s, annealing at variable temperatures for 30 s (set at 60 ˚C and decreasing by 0.5 ˚C per cycle) and extension at 72 ˚C for 1 min; 2) 25 cycles of 94˚C for 30 s, 50 ˚C for 30 s, and 72˚C for 1 min, followed by a final extension of

72 ˚C for 7 min. The PCR products, after visualized on agarose gel, were purified using the NZYGelpure kit (from NZYTech, Lda) and sequencing was done commercially (Macrogen Inc.).

Electropherograms were inspected using Genestudio V.2.2.0.0 (www.genestudio.com) and sequences were cropped to the minimal region of overlap for all individuals using the same software. The 21 *Bactrocera oleae* full mitochondrial genomes made available by Nardi and co-workers [24] were aligned with our sequences and the same regions of overlap were extracted. All alignments were made using the Muscle tool implemented in MEGA X [25]. The amplified mitochondrial DNA segments and the database obtained ones were concatenated per individual. Median-joining networks [26] were calculated using the PopART [27] software.

## nDNA genotyping

Ten microsatellite loci were used in the analysis (Table 2) [28]. PCR amplification was performed in a total volume of 15 μl, with.5 mM $MgCl^2$, 0.5 mM dNTPs, 0.04 U/ml Taq DNA polymerase and 0.5 pmol/μl of each locus-specific primer with one of the primers in pair elongated for M13(-21) universal sequence (Schuelke 2000), 0.25 lM of M13(-21) universal primer labelled with dyes 6-FAM, VIC, PET or NED (Applied Biosys- tems, Foster City, CA, USA), 0.375 unit of Taq DNA polymerase (Fermentas, Vilnius, Lithuania) and 25 ng of genomic DNA. Fragment analysis by capillary electrophoresis was performed commercially at STAB VIDA Lda. Microsatellite genotypes were visualized and manually controlled with GeneMapper version 4.1 software (Applied Biosystems). GenAlEx version 6.5 [29] was used to determine the following parameters of genetic variability: number of alleles (n), number of effective alleles (ne), observed heterozygosity (Ho), expected heterozygosity (He) and Unbiased Expected heterozygosity (uHe). POPGENE version 1.31 [30] was used to calculate deviations from Hardy–Weinberg equilibrium with likelihood ratio (G2 test) and for determination of the Shannon's Information Index (I) and fixation index (Fst) and inbreeding coefficient (Fis) and gene flow (Nm) between populations. The genetic distances were estimated on the same

**Table 2. Microsatellite [28] loci motif, accession numbers and primer used.**

| Locus name | Motif | Acc. Number | Primer sequence |
|---|---|---|---|
| Boms5 | $(CA)_{10}TA(CA)_2 (CA)_9$ | EU489749 | F: TGTAAAACGACGGCCAGTTCTCGCCCCAATTACCAC<br>R: GAATTTTGGCAACATCCAAGC |
| Boms8 | $(CA)_7CG(CA)_5$ | EU489752 | F: TGTAAAACGACGGCCAGT TGACATACATGCCTTCATTCAC<br>R: CAGAAAAGCTTAAAACTAGCGG |
| Boms18 | $(CA)_{13}$ | AF467828 | F: TGTAAAACGACGGCCAGTGCCATGAATGCAGACCAC<br>R: CCTATTCAAATGCACGCAAAAC |
| Boms21 | $GTGG(GT)_{13}ATGT$ | AF467827 | F: TGTAAAACGACGGCCAGTTCGCCTCTTACCTCACAACC<br>R: ACCATCCTTAGTCAGCACAGTC |
| Boms25 | $(GT)_{12}$ | AF467826 | F: TGTAAAACGACGGCCAGT TGGAATGCGCTATTTTGTTG<br>R: ACTCGTATATACGTACATGG |
| Boms30 | $(GT)_{17}$ | AF467823 | F: TGTAAAACGACGGCCAGT CTGACTTCTTGCTTTACACG<br>R: CAGCTTATCTGCTTTAAGTGC |
| Boms32 | $(CA)_{14}$ | EU489765 | F: TGTAAAACGACGGCCAGTTGTATGTATTTGTGCGTCG<br>R: GCTTAGACCATTTGCTCC |
| Boms34 | $(CA)_3CTA(CA)_8$ | EU489767 | F: TGTAAAACGACGGCCAGTACGCCGCACACTTCTTAAAC<br>R: CACCCAACTTTTGTAGTTTCC |
| Boms58 | $A_6CA_3GCA_6TA_5CA_5$ | EU489782 | F: TGTAAAACGACGGCCAGTAGTTGGACGCGCACATATC<br>R: AGCGCGTACGAGCTTTAGC |
| Boms59 | $TGTA(TG)_{10}$ | DQ078250 | F: TGTAAAACGACGGCCAGTAGCGCTTACATAAATATAGCTAC<br>R: TCCCCGTAAAGCCATAAAGTC |

software using Nei's (1978) unbiased genetic distance coefficient, and the dendrogram was constructed based on the unweighted pair-group arithmetic average (UPGMA) method and visualized in Mega X [25]. STRUCTURE 2.3.4 [31] was used to classify the individuals into a set number of clusters (K). Using the admixture ancestry model and correlated allele frequency model different values for K, from one to ten, were tested running the analysis 10 times for each cluster (each run consisted of a burn-in period of 50 000 and 100 000 Markov chain Monte Carlo (MCMC) repetitions after the initial burn-in). The most appropriate cluster number was selected using the method reported by [32].

### *Ca*. Erwinia dacicola 16S sequences

For the selective amplification of a fragment sequence of the 16S recombinant deoxyribonucleic acid (rDNA) of *Ca*. Erwinia dacicola, a specific primer (EdF1) was paired with 1507R for PCR as previously described [11]. PCR reactions were conducted using 1 μl of the extracted DNA in a standard 25 μl reaction, with 0.5 pmol/μl of each primer, 1.5 mM $MgCl^2$, 0.5 mM dNTPs and 0.04 U/ml Taq DNA polymerase with the following cycle protocol: initial denaturation at 94 ˚C for 5 min, followed by 30 cycles of denaturation at 94 ˚C for 30 s, annealing for 30 s at 55 ˚C and extension at 72 ˚C for 30 s, followed by a final extension of 72 ˚C for 10 min. The PCR products, after visualized on agarose gel, were purified using the NZYGelpure kit (from NZYTech, Lda) and sequencing was done commercially (Macrogen Inc.). Electropherograms were inspected using Genestudio (www.genestudio.com). The evolutionary history was inferred by Maximum Likelihood and using the Hasegawa-Kishino-Yano model as selected model of evolution. Both analyses were performed in Mega X [25].

## Results

From the sections of mtDNA sequenced, we retrieved a total of 3626 bp and 87 polymorphic sites (S2 Table) and Table 3 further summarizes nucleotide data at the four segments analysed, including Tajima's Neutrality Test that suggest that the DNA sequences studied are evolving under a non-random process, with many low frequency alleles causing low average divergence —characteristic of population expansion.

The Median-joining work representing relationships among mitochondrial haplotypes of olive fly is represented in Fig 1.

All ten microsatellite markers were polymorphic, revealing a total of 54 alleles with the parameters of genetic variability being presented on Table 4.

**Table 3. Nucleotide frequencies and polymorphisms in *Bactrocera oleae* (S = number of segregating sites, ps = S/m, Θ = ps/a1, π = nucleotide diversity, and D is the Tajima test statistic).**

| gene | Nucleotide Frequencies (%) | | | | | Tajima's Neutrality Test | | | | |
|---|---|---|---|---|---|---|---|---|---|---|
| | T | C | A | G | Total[2] | S | ps | Θ | π | D |
| ND2 | 36.5 | 18.4 | 35.1 | 9.9 | 1005.9 | 8 | 0.009 | 0.002 | 0.001 | -1.537 |
| Cox | 33.6 | 20.6 | 28.9 | 16.9 | 605.1 | 14 | 0.023 | 0.005 | 0.002 | -1.490 |
| CLC [1] | 33.7 | 21.4 | 32.3 | 12.6 | 989.4 | 39 | 0.039 | 0.008 | 0.003 | -2.018 |
| ND4 | 26.9 | 16.7 | 49.1 | 7.3 | 870.7 | 12 | 0.012 | 0.002 | 0.001 | -1.154 |
| Total | 32.8 | 19.2 | 36.7 | 11.3 | 3471.1 | 88 | 0.024 | 0.005 | 0.002 | -1.838 |

[1] fragment comprising part of COX1, tRNA-Leu (UUR) and COX2;

[2] average number of total nucleotides.

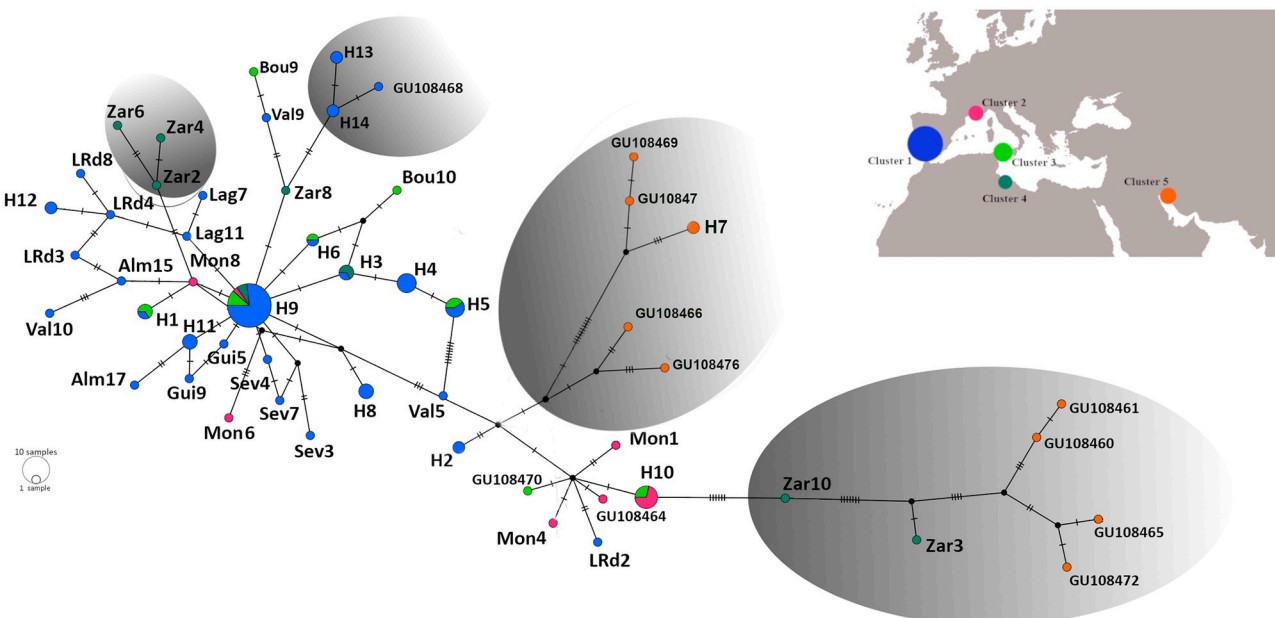

**Fig 1. Median-joining network representing relationships among the haplotypes (S2 Table) from the analysed segments of mitochondrial DNA of the olive fruit fly (*Bactrocera oleae*).** Circles represent haplotypes and the size of the circles is proportional to the frequency of the haplotype (black circles represent unobserved intermediate haplotypes and the length of the connections is proportional to the number of mutational steps that separate the haplotypes; these shared haplotypes are described on S3 Table). Colours correspond to the geographical cluster represented in the map. The grey coloured ovals correspond to maximum likelihood clades with a bootstrap value higher than 70% (in 1000 replicates; S1 Fig).

Genetic distances among samples from all the locations were measured according to Nei's unbiased genetic distance and the UPGMA dendrogram constructed suggests the grouping of data in two different clusters (Fig 2) albeit not related with a clear geographical pattern.

Whereas the UPGMA dendogram by definition depicts a dichotomous structure, this should be interpreted only has similarities between the haplotypes as one cluster seems the most likely number of clusters (K) according to the STRUCTURE analysis (Fig 3).

**Table 4. Parameters of genetic variability: Number of alleles (n), number of effective alleles (ne), observed heterozygosity (Ho), expected heterozygosity (He), Unbiased Expected heterozygosity (uHe), information index (I), Hardy–Weinberg equilibrium (HWE), fixation index (Fst) and inbreeding coefficient (FIS) and gene flow (Nm) among sampling sites.**

|  | N | ne | Ho | He | uHe | I | HWE (p) | Fst | Fis | Nm |
|---|---|---|---|---|---|---|---|---|---|---|
| Boms5 | 5 | 1.762 | 0.398 | 0.418 | 0.443 | 0.814 | 0.07 | 0.035 | 0.048 | 6.971 |
| Boms8 | 6 | 1.591 | 0.368 | 0.342 | 0.362 | 0.649 | 0.92 | 0.078 | -0.076 | 2.972 |
| Boms18 | 5 | 3.023 | 0.784 | 0.658 | 0.696 | 1.187 | 0.47 | 0.107 | -0.191 | 2.084 |
| Boms21 | 6 | 2.479 | 0.609 | 0.560 | 0.599 | 1.062 | 0.62 | 0.088 | -0.087 | 2.577 |
| Boms25 | 6 | 2.575 | 0.607 | 0.602 | 0.638 | 1.073 | 0.66 | 0.047 | -0.010 | 5.034 |
| Boms30 | 5 | 2.414 | 0.541 | 0.570 | 0.605 | 0.991 | 0.00 | 0.093 | 0.051 | 2.441 |
| Boms32 | 8 | 4.089 | 0.852 | 0.748 | 0.795 | 1.537 | 0.40 | 0.049 | -0.139 | 4.845 |
| Boms34 | 4 | 2.250 | 0.524 | 0.543 | 0.576 | 0.932 | 0.46 | 0.067 | 0.035 | 3.494 |
| Boms58 | 4 | 2.589 | 0.609 | 0.592 | 0.627 | 1.071 | 0.59 | 0.069 | -0.028 | 3.396 |
| Boms59 | 5 | 2.039 | 0.379 | 0.487 | 0.514 | 0.874 | 0.01 | 0.071 | 0.222 | 3.266 |

*p value ($P < 0.05$).

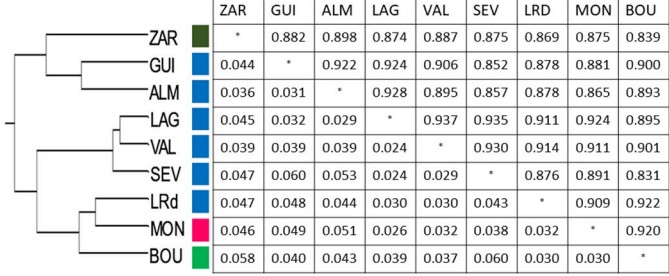

| | ZAR | GUI | ALM | LAG | VAL | SEV | LRD | MON | BOU |
|---|---|---|---|---|---|---|---|---|---|
| ZAR | * | 0.882 | 0.898 | 0.874 | 0.887 | 0.875 | 0.869 | 0.875 | 0.839 |
| GUI | 0.044 | * | 0.922 | 0.924 | 0.906 | 0.852 | 0.878 | 0.881 | 0.900 |
| ALM | 0.036 | 0.031 | * | 0.928 | 0.895 | 0.857 | 0.878 | 0.865 | 0.893 |
| LAG | 0.045 | 0.032 | 0.029 | * | 0.937 | 0.935 | 0.911 | 0.924 | 0.895 |
| VAL | 0.039 | 0.039 | 0.039 | 0.024 | * | 0.930 | 0.914 | 0.911 | 0.901 |
| SEV | 0.047 | 0.060 | 0.053 | 0.024 | 0.029 | * | 0.876 | 0.891 | 0.831 |
| LRd | 0.047 | 0.048 | 0.044 | 0.030 | 0.030 | 0.043 | * | 0.909 | 0.922 |
| MON | 0.046 | 0.049 | 0.051 | 0.026 | 0.032 | 0.038 | 0.032 | * | 0.920 |
| BOU | 0.058 | 0.040 | 0.043 | 0.039 | 0.037 | 0.060 | 0.030 | 0.030 | * |

**Fig 2. UPGMA dendrogram of the different sampled populations based on Nei's unbiased genetic distance; the color codes correspond to cluster colors on Fig 1.** The associated table shows Nei's genetic identity values (upper diagonal) and pairwise population FST values (below diagonal).

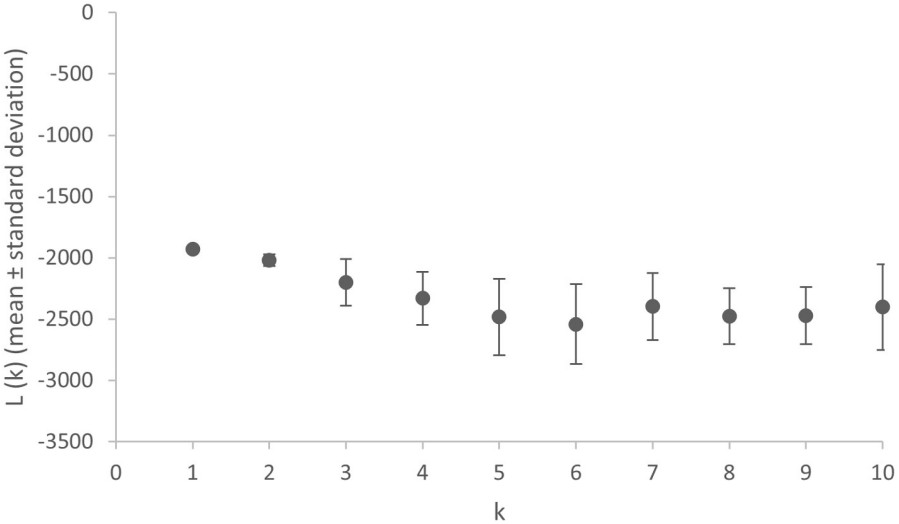

**Fig 3. Plot of mean ln P(X|K) over 10 independent runs for each K (K = 1–10) suggests a higher likelihood of one cluster only.**

The 16S fragment sequenced of the symbiont *Ca*. Erwinia dacicola specimens showed only 20 parsimonious informative sites leading to a reconstructed phylogeny (Fig 4) where the two known haplotypes -htA and htB- are in a clade (two SNPs only) and represented by 69 and 14 sequences respectively (S3 Table). Two new haplotypes, htC and htD with only 3 and 4 representatives each were found exclusively in Zarzis (htC) or Boufiche (htD) populations (note that these sequences were confirmed by two independent reactions of PCR and sanger sequencing). htC diverges from htA at 7 SNPs (and from htB at 9 SNPs), but also diverges from htD at 11 SNPs. The biggest pairwise differences are found between htD and htB (20 SNPs compared to the 18 SNPs with htA) (S1 File for sequence details).

## Discussion

The so far established structure of *B. oleae* in the Mediterranean Basin points to the existence of three clusters, with the olive fly populations in Iberia and the Levant differentiating from the ones of the Italic peninsula, albeit with intermixing [8,24]. The present work focused on the Iberian cluster and hypothesised that the Tunisian samples would fall into a differentiated cluster due to the proximity to the Italic peninsula cluster. The data obtained does not allow to

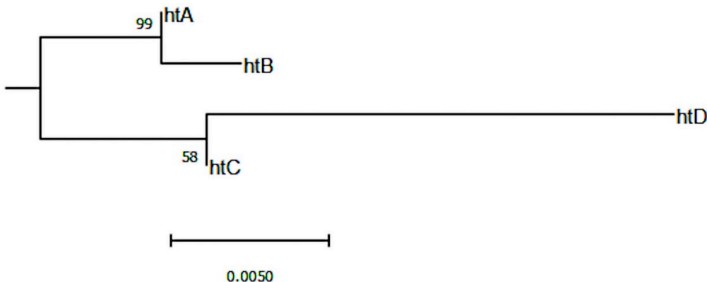

**Fig 4. Phylogenetic reconstruction of the *ca*. Erwinia dacicola fragment sequence (814 bp after quality trimming and alignment) of the 16S recombinant deoxyribonucleic acid (rDNA) gene.** Three other *Erwinia* sequences (KC139453, *E. persicina*; KT766070 *E. aphidicola*; JX867759, *E. piriflorinigrans*) were used to root the phylogeny (not shown). The tree with the highest log likelihood (-1304.66) is shown together with the percentage of trees in which the associated taxa clustered together (next to the branches, 1000 replicates). The selected Hasegawa-Kishino-Yano model with a discrete Gamma distribution was used to model evolutionary rate differences among sites (+G, parameter = 0.05). The tree is drawn to scale, with branch lengths measured in the number of substitutions per site.

confirm or dismiss the hypothesis, and a more detailed work with a higher number of samples and locations is needed. If something, the data on the fly suggests that population in this area is better seen as single cluster with no clear differentiation between populations. If for the mtDNA fragments the number of samples per population is not critical, it is well below the advisable number for microsatellite analyses (between 25 to 30 individuals seem to be needed for a reliable estimation of allele frequencies; [33]). Unfortunately, sampling relied on number of emerging adults from collected olive fruits and a high number of fruits render no insects, either by not having been infected or by death of the larvae due to storage and/or shipping conditions.

However, and when looking at the obligate symbiont *Ca*. Erwinia dacicola, two new 16S haplotypes were found exclusively in the populations from Tunisia. The olive fruit fly exhibits a strongest dependence of its endosymbiont *Ca*. E. dacicola, as it on its own does not possess the enzymatic systems required for feeding on unripped olive fruit. This actually means that the fly is a holobiont [34], better seen as a group of genetically different entities subjected to natural selection (at the level of holobiont and hologenome rather than individuals or genomes [35]). This degree of dependence associated to a vertical mode of transmission strengthens the integrity of the holobiont and stresses the importance of looking at the genetic diversity of the obligate symbiont for our understanding of the host population dynamics.

The geographical data on the *Ca*. E. dacicola is still scarce. The present work duplicates the known number of 16S haplotypes. These new haplotypes seem to make a sister group with the two already described 16S haplotypes [36] (Fig 4), and so far they were only found south of western Mediterranean Basin. Two hypotheses could be put forward explaining these two sister groups: 1) a single event of symbiont acquisition followed by symbiont divergence through evolution, driven by geographical or functional determinants; 2) two events of symbiont acquisition, either unrelated or due to symbiont loss and de novo acquisition through a horizontal transmission event. The limited geographical data available does not allow to disentangle between the raised hypotheses.

Following the principle that "the simplest explanation is usually the best one", a single event of symbiont acquisition is the most likely hypothesis. In vertically transmitted symbionts there is a reduction in the purifying activity of natural selection [37,38] and the symbiont population is subjected to transmission bottlenecks, both with consequences on the symbiont genome. Theory predicts that strictly vertical transmitted symbionts are associated with high rates of amino acid substitution and the resulting reduced nucleotide base composition does not

favour guanidine (G) and cytosine (C) (reviewed in [39]). Likewise, they show gene inactivation and loss, and ultimately a reduced genome ([39–41]). However, the data so far on *Ca.* Erwinia dacicola shows that both the genome size and GC content are similar to those of free-living bacteria than to those of other intracellular bacteria found in other insects [42,43]. These features are usually associated with facultative symbionts or with symbionts that only recently became obligate [44]. From what we know on this symbiosis, none of those special cases seem to apply. First, *Ca.* Erwinia dacicola seems to be an obligate symbiont, as it is believed that it allows the insect to overcome olive-plant secondary metabolites (and in particular the oleuropein) [6] and it is present in all life stages of wild olive fruit flies, being thus maintained through natural changes in diet and host metamorphosis [45]. Secondly, the olive fruit fly differentiation in the Mediterranean seems to be connected with the post-glacial recolonization of wild olives in the area [46], suggesting that symbiont acquisition is also not recent as the fly seems to be long exploiting this olive fruit resource. In such a scenario, the 'Erwinia' transition into an obligate symbiont probably occurred far into the evolutionary history of the olive fruit fly, before the tree domestication begun. Note needs to be taken here, as experimental data have shown that the olive fruit fly can survive and reach adulthood on ripped fruit without the obligate symbiont [6]. This raises the alternative hypothesis of a recent shift of this 'Erwinia' species into an obligate, vertical transmitted symbiont, probably following the domestication and human-mediated spread of this culture. On the other hand, this bacterium was found to transition between intracellular and extracellular lifestyles during specific stages of the host's life cycle which together with the need to cope with a somewhat changing environment during the development of its polyphagous, holometabolous host, can be the reasons for the genomic similarity with free-living bacteria [11]. Also needing to be acknowledge is the possibility of DNA exchange with other transient gut bacteria (e.g., free living *Enterobacter* spp.) during the endosymbiont extracellular life stage, where they might coexist in the gut environment. Either way, a single transition to an obligate symbiont that became transmitted 100% vertically and uniparental (via the female) should lead to a structure genetically uniform in a clonal manner (clonality defined as in [47], the balance between vertical and horizontal gene inheritance amongst bacteria) purging genetic diversity from the population. Yet we now found a different *Ca.* E. dacicola 16S clade which on average diverges about 2% from the previously 16S known haplotypes. Only a study designed to disentangle the life history of these lineages will be able to distinguish between the different hypotheses, and in particularly between eventual recombination events, episodes of horizontal transmission of the symbiont or a 'de novo' acquisition event. An extracellular lifestyle of this symbiotic bacteria could indeed potentiate recombination events via coprophagy and/or the trophallaxis behaviour, as horizontal transfer of symbionts was observed in controlled conditions via cohabitation of symbiont-free lab populations and wild flies [48].

Nonetheless, we are dealing with a tripartite symbiosis with high specificities, and to understand olive fruit fly population diversity and dispersion there is a need to look at the olive tree population, its cultivars' diversity and specificities, and also to the symbiont that enables the fly to so efficiently and uniquely exploit the fruit of the olive tree. All data suggest for high levels of intermixing of the fly populations, including a fast spreading of genes with fitness advantage (this ample gene flow is clear from the studies on the alleles associated with OPs resistance; [46,49,50]). The association between *Bactrocera oleae* and its obligate symbiont *Candidatus* Erwinia dacicola is key to success of the fly, enabling several generations per year as unripped fruit can be used successfully. Understanding this relation will aid in understanding pest dispersion and dynamics and in the search of alternative and sustainable pest management methods (symbiosis based; [51,52]).

## Supporting information

**S1 Fig. *Bactrocera oleae* maximum likelihood reconstruction.**
(DOCX)

**S1 Table. Primers used to amplify *Bactrocera oleae* mitochondrial DNA segments analysed in this study.**
(XLS)

**S2 Table. Sequence variability having the *Bactrocera oleae* mitochondrion, complete genome AY210702 as reference.**
(XLS)

**S3 Table. Mitochondrial haplotypes shared by different specimens.**
(XLS)

**S1 File. 16S variable sites and haplotype inference.**
(XLS)

## Acknowledgments

I would like to deeply thank Imen Blibech for the samples from Tunisia, Raquel Garcia for the Montpellier samples, Juan Olivares and the Agropecuaria Vallefrío Nueva S.L. in La Roda (Albacete) for the Spanish samples and Fernando T. Rei for the Portuguese samples. Without their contribution, the present work would not have been possible.

## Author Contributions

**Conceptualization:** Tânia Nobre.

**Formal analysis:** Tânia Nobre.

**Funding acquisition:** Tânia Nobre.

**Investigation:** Tânia Nobre.

**Methodology:** Tânia Nobre.

**Project administration:** Tânia Nobre.

**Writing – original draft:** Tânia Nobre.

**Writing – review & editing:** Tânia Nobre.

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
