## [Decision Letter · Decision Letter 0]

3 Jun 2021

PONE-D-21-11736

Olive fruit fly and its obligate symbiont Candidatus Erwinia dacicola: two new symbiont haplotypes in the Mediterranean basin

PLOS ONE

Dear Dr. Nobre,

Thank you for submitting your manuscript to PLOS ONE. After careful consideration, we feel that it has merit but does not fully meet PLOS ONE’s publication criteria as it currently stands. Therefore, we invite you to submit a revised version of the manuscript that addresses the points raised during the review process.

Comments of both reviews should be carefully considered. Additional effort should be invested to addresses several minor editorial points in both text and the references cited that need your immediate and careful attention. I have included an annotated copy of your ms with several marks. 

We look forward to receiving your revised manuscript.

Kind regards,

Nikos T Papadopoulos

Academic Editor

PLOS ONE

Journal Requirements:

4. We note that Figure 1 in your submission contains map images which may be copyrighted.

We require you to either (a) present written permission from the copyright holder to publish this figure specifically under the CC BY 4.0 license, or (b) remove the figure from your submission:

b. If you are unable to obtain permission from the original copyright holder to publish this figure under the CC BY 4.0 license or if the copyright holder’s requirements are incompatible with the CC BY 4.0 license, please either i) remove the figure or ii) supply a replacement figure that complies with the CC BY 4.0 license. Please check copyright information on all replacement figures and update the figure caption with source information. If applicable, please specify in the figure caption text when a figure is similar but not identical to the original image and is therefore for illustrative purposes only.

Reviewers' comments:

Reviewer's Responses to Questions

**Comments to the Author**

1. Is the manuscript technically sound, and do the data support the conclusions?

Reviewer #1: Yes

Reviewer #2: Yes

2. Has the statistical analysis been performed appropriately and rigorously? 

Reviewer #1: Yes

Reviewer #2: I Don't Know

3. Have the authors made all data underlying the findings in their manuscript fully available?

Reviewer #1: Yes

Reviewer #2: No

4. Is the manuscript presented in an intelligible fashion and written in standard English?

Reviewer #1: Yes

Reviewer #2: Yes

5. Review Comments to the Author

Reviewer #1: The author followed a holistic approach combining data from both the host and the endosymbiont. Interestingly two new 16S haplotypes regarding the Candidatus Erwnia dacicola were detected for the Tunisian olive fly populations, which is the major finding of the manuscript, adding valuable data in the existing findings regarding the endosymbiont genetic variability.

There is also an interesting overall commentary regarding the olive fly population diversity and coevolution with the symbiont.

Generally, the manuscript is well written and structured employing a wide list of genomic tools and approaches. There is a main disadvantage regarding the microsatellites part with a restricted number of flies used for the analysis in order to have a reliable estimation. Author has explained the reason behind this and therefore assumptions based on microsatellites throughout the text are minimum.

There is a short list of minor issues that I would like to be addressed through a revised version of the manuscript:

A supplemental list with the primers/probes used would be beneficial for the reader.

Please do not use “e.g.” for citing references within the text.

Line 87-88: “These changes will lead to species specific dynamics with impact on the interaction of the olive and its obligate olive fruit fly”. This is a strong statement and it should be followed by a list of relevant references.

Line 249: “following the parsimony” please explain the meaning here and if necessary add references to support it.

Line 270: correct “this bacteria was found”

Line 287: “uniquely exploit this resource” where do you refer by “resource”?

Line 287: please correct “all data suggests” to “all data suggest”.

Reviewer #2: PONE-D-21-11736: Olive fruit fly and its obligate symbiont Candidatus Erwinia dacicola: two new symbiont haplotypes in the Mediterranean basin

This paper identifies two new haplotypes of the olive fly symbiont, Ca. Erwinia dacicola, found to be uniquely associated with flies from Tunisia, and significantly different from previously described haplotypes associated with samples from Europe. These findings contrast analyses based on genomic and mitochondrial markers suggesting that the populations sampled from Africa and Europe cluster with no clear association to their geographic location. The main conclusion is that since the symbiont is vertically transmitted and essential to its host the new bacterial linages may contribute to understanding the dispersion patterns and population diversity of the fly.

The paper is well written but please note that there are many remaining typos, syntax and parsing issues which need to be attended. From the perspective of my field of expertise I have only a few comments, mostly concerning topics addressed in the results and discussion.

Please make sure that the sequencing data underlying the findings will be made publically available (I couldn’t find a note on this in the manuscript).

Introduction:

Page 4, Lines 83-84 and 222-224 (discussion): please state the hypotheses of this study in the introduction as well.

Results:

1. Figure 1 depicts the median-joining network of haplotypes derived from mitochondrial DNA analysis. In addition to the populations sampled in this study (9 locations), other flies seem to have been included in the analysis (e.g. H, GU, orange coded), which do not seem to be mentioned in the text. Please include a description of these samples. Mont = MON?

2. In Figure 2 the color codes for Montpelier and Bouficha seem to have been mixed (they do not match those in figure 1). Should there be a representation of the orange-coded cluster as well (as in figure 1)?

3. The STRUCTURE analysis and figure 3: this is a bit confusing. Should the microsatellite-based dendogram depict two clusters (as currently presented) or only one cluster? Please include a short clarification for readers which are unfamiliar with the technicalities of this specific test.

Discussion:

The discussion addresses several issues but is currently presented as one continuous mass of text. I suggest to divide the text into clear sections complying with the addressed subjects.

Lines 245 – 248: according to file S3, flies sampled from Tunisia (and other locations as well) can be associated with either one of the 4 identified symbiont haplotypes. Two points which might be considered to be included in the discussion: (1) Can the above be congruent with these two hypotheses?, and (2) can certain fly genotypes (e.g. mitochondrial haplotypes) be more prone to be associated with a certain bacterial haplotype?

Line 268: I would separate the evolution of Erwinia into an obligate symbiont from the domestication of the olive. The transition into an obligate symbiont probably occurred far into the evolutionary history of the fly, and is not associated with recent human activity.

Lines 269 – 273: another possible explanation contributing to retention of genome size of the Erwinia symbiont: it is a gut bacterium, and as such it remains exposed to transient bacteria which are ingested with the diet (probably mainly at the adult stage), and thus, to horizontal gene transfer. The composition of the gut microbiome of this fly suggest that bacteria other than the Erwinia symbiont (e.g. free living Enterobacter spp.) are intermittently associated with the fly and may contribute to DNA exchange.

6. PLOS authors have the option to publish the peer review history of their article (what does this mean?). If published, this will include your full peer review and any attached files.

Reviewer #1: No

Reviewer #2: No

---

## [Author Response · Author response to Decision Letter 0]

9 Jun 2021

Reviewer #1: The author followed a holistic approach combining data from both the host and the endosymbiont. Interestingly two new 16S haplotypes regarding the Candidatus Erwnia dacicola were detected for the Tunisian olive fly populations, which is the major finding of the manuscript, adding valuable data in the existing findings regarding the endosymbiont genetic variability. There is also an interesting overall commentary regarding the olive fly population diversity and coevolution with the symbiont.

Generally, the manuscript is well written and structured employing a wide list of genomic tools and approaches. There is a main disadvantage regarding the microsatellites part with a restricted number of flies used for the analysis in order to have a reliable estimation. Author has explained the reason behind this and therefore assumptions based on microsatellites throughout the text are minimum.

There is a short list of minor issues that I would like to be addressed through a revised version of the manuscript:

A supplemental list with the primers/probes used would be beneficial for the reader.

This list was added as supplemental file.

Please do not use “e.g.” for citing references within the text.

Removed.

Line 87-88: “These changes will lead to species specific dynamics with impact on the interaction of the olive and its obligate olive fruit fly”. This is a strong statement and it should be followed by a list of relevant references.

Three relevant references were added to the text, that show that climate change will impact differently different insect species. 

Line 249: “following the parsimony” please explain the meaning here and if necessary add references to support it.

The sentence was changed for better accuracy. What was meant as the principle of parsimony was the principle that the simplest explanation is usually the best one.

Line 270: correct “this bacteria was found”

Corrected

Line 287: “uniquely exploit this resource” where do you refer by “resource”?

By resource I meant the fruit of the olive tree. Sentence was changed accordingly

Line 287: please correct “all data suggests” to “all data suggest”.

Corrected

 

Reviewer #2: PONE-D-21-11736: Olive fruit fly and its obligate symbiont Candidatus Erwinia dacicola: two new symbiont haplotypes in the Mediterranean basin

This paper identifies two new haplotypes of the olive fly symbiont, Ca. Erwinia dacicola, found to be uniquely associated with flies from Tunisia, and significantly different from previously described haplotypes associated with samples from Europe. These findings contrast analyses based on genomic and mitochondrial markers suggesting that the populations sampled from Africa and Europe cluster with no clear association to their geographic location. The main conclusion is that since the symbiont is vertically transmitted and essential to its host the new bacterial linages may contribute to understanding the dispersion patterns and population diversity of the fly.

The paper is well written but please note that there are many remaining typos, syntax and parsing issues which need to be attended. From the perspective of my field of expertise I have only a few comments, mostly concerning topics addressed in the results and discussion. Please make sure that the sequencing data underlying the findings will be made publically available (I couldn’t find a note on this in the manuscript).

The sequencing data on the olive fruit fly are directly available from the S1 Table as it shows all the nucleotide variability in relation to Bactrocera oleae mitochondrion, complete genome AY210702 as reference. The four Ca. Erwinia dacicola 16S haplotypes are also derived from the data made available on S3 File and were also submitted to GenBank under the codes htA: MW888710, htD: MW888711, htB: MW888712 and htC: MW888713.

Introduction:

Page 4, Lines 83-84 and 222-224 (discussion): please state the hypotheses of this study in the introduction as well.

Thank you. This was done and can now be read on Lines 79-80.

Results:

1. Figure 1 depicts the median-joining network of haplotypes derived from mitochondrial DNA analysis. In addition to the populations sampled in this study (9 locations), other flies seem to have been included in the analysis (e.g. H, GU, orange coded), which do not seem to be mentioned in the text. Please include a description of these samples. Mont = MON?

Thanks for the comment, which highlights that the information might not have been clear. The other sequences included are referred in the methodology L-119 to L-121 and these are represented by the GenBank codes and H# represents a haplotype shared by different samples. This information is indeed only presented in (S1 Table) and the haplotypes needed to be derived. Therefore, I added a S2 Table, facilitating the interpretation. A sentence is added to the legend now, for clarification. Mont = MON (now corrected).

2. In Figure 2 the color codes for Montpelier and Bouficha seem to have been mixed (they do not match those in figure 1). Should there be a representation of the orange-coded cluster as well (as in figure 1)?

You are absolutely correct as to the color codes of MON and BOU. Thank you. They are now corrected. Unfortunately, no representative of orange-coded cluster from figure 1 was available in the sampled specimens (they all corresponded to sequences coming from previous study) and therefore they could not be included in the microsatellite analyses. 

3. The STRUCTURE analysis and figure 3: this is a bit confusing. Should the microsatellite-based dendogram depict two clusters (as currently presented) or only one cluster? Please include a short clarification for readers which are unfamiliar with the technicalities of this specific test.

The UPGMA dendogram by definition depicts a dichotomous structure whereas STRUCTURE infers the most likely number of clusters for a given data set using a Bayesian approach. A sentence was added to aid the reader, as suggested.

Discussion:

The discussion addresses several issues but is currently presented as one continuous mass of text. I suggest to divide the text into clear sections complying with the addressed subjects.

Thank you for the suggestion. I did struggle with finding clear sections complying with the results addressed because my original effort was exactly the opposite: I have attempted to give a more holistic perspective to the discussion by combining data from both the host and the endosymbiont. I do believe that this is one of the strong points of the manuscript, also recognized by reviewer one. Therefore, I decided to stay with my original approach.

Lines 245 – 248: according to file S3, flies sampled from Tunisia (and other locations as well) can be associated with either one of the 4 identified symbiont haplotypes. Two points which might be considered to be included in the discussion: (1) Can the above be congruent with these two hypotheses?, and (2) can certain fly genotypes (e.g. mitochondrial haplotypes) be more prone to be associated with a certain bacterial haplotype?

No doubt are the points raised interesting and relevant. Unfortunately, the scarce geographical data available does not allow to dismiss none of the two hypotheses raised in the main text. Also, the interesting possibility of high host-symbiont haplotype specificity is not possible to address with the current data. However, as soon as more data is available, I believe these suggestions can and should be discussed. A sentence was added to the discussion to allocate this comment and the data limitation.

Line 268: I would separate the evolution of Erwinia into an obligate symbiont from the domestication of the olive. The transition into an obligate symbiont probably occurred far into the evolutionary history of the fly, and is not associated with recent human activity.

This was presented in the discussion as the main hypothesis, in lines 264-267. However, it seems that it was not clear enough and therefore a sentence was added for clarification. Also further on the sentence was re-written to accommodate the comment and becoming clearer. 

Lines 269 – 273: another possible explanation contributing to retention of genome size of the Erwinia symbiont: it is a gut bacterium, and as such it remains exposed to transient bacteria which are ingested with the diet (probably mainly at the adult stage), and thus, to horizontal gene transfer. The composition of the gut microbiome of this fly suggest that bacteria other than the Erwinia symbiont (e.g. free living Enterobacter spp.) are intermittently associated with the fly and may contribute to DNA exchange.

This hypothesis is now stated in a clearer way (Line 276-279).

---

## [Decision Letter · Decision Letter 1]

4 Aug 2021

Olive fruit fly and its obligate symbiont Candidatus Erwinia dacicola: two new symbiont haplotypes in the Mediterranean basin

PONE-D-21-11736R1

Dear Dr. Nobre,

We’re pleased to inform you that your manuscript has been judged scientifically suitable for publication and will be formally accepted for publication once it meets all outstanding technical requirements.

Kind regards,

Nikos T Papadopoulos

Academic Editor

PLOS ONE

Additional Editor Comments (optional):

There is need for careful editing especially in references cited. Italics should be used for all scientific names provided

Reviewers' comments:

Reviewer's Responses to Questions

**Comments to the Author**

1. If the authors have adequately addressed your comments raised in a previous round of review and you feel that this manuscript is now acceptable for publication, you may indicate that here to bypass the “Comments to the Author” section, enter your conflict of interest statement in the “Confidential to Editor” section, and submit your "Accept" recommendation.

Reviewer #1: All comments have been addressed

Reviewer #2: All comments have been addressed

2. Is the manuscript technically sound, and do the data support the conclusions?

Reviewer #1: Yes

Reviewer #2: (No Response)

3. Has the statistical analysis been performed appropriately and rigorously? 

Reviewer #1: Yes

Reviewer #2: (No Response)

4. Have the authors made all data underlying the findings in their manuscript fully available?

Reviewer #1: Yes

Reviewer #2: (No Response)

5. Is the manuscript presented in an intelligible fashion and written in standard English?

Reviewer #1: Yes

Reviewer #2: (No Response)

6. Review Comments to the Author

Reviewer #1: The author addressed all my comments therefore I fully support the publication of the current manuscript.

Reviewer #2: (No Response)

7. PLOS authors have the option to publish the peer review history of their article (what does this mean?). If published, this will include your full peer review and any attached files.

Reviewer #1: No

Reviewer #2: No

---

## [Editor Report · Acceptance letter]

20 Aug 2021

PONE-D-21-11736R1 

Olive fruit fly and its obligate symbiont *Candidatus* Erwinia dacicola: two new symbiont haplotypes in the Mediterranean basin 

Dear Dr. Nobre:

I'm pleased to inform you that your manuscript has been deemed suitable for publication in PLOS ONE. Congratulations! Your manuscript is now with our production department. 

Kind regards, 

on behalf of

Dr. Nikos T Papadopoulos 

Academic Editor

PLOS ONE